# Maternal Variability of Amplitudes of Frequency Fluctuations Is Related to the Progressive Self–Other Transposition Group Intervention in Autistic Children

**DOI:** 10.3390/brainsci13050774

**Published:** 2023-05-08

**Authors:** Jianxin Zhang, Xiaorong Guo, Weiguo Zhang, Dianzhi Liu, Peiqi Chen, Yuqing Zhang, Xiaoyuan Ru

**Affiliations:** 1The Autism Research Center, Soochow University, Suzhou 215123, China; 8201807147@jiangnan.edu.cn; 2School of Education, Jiangnan University, Wuxi 214122, China; 3Department of Radiology, Dushu Lake Public Hospital Affiliated to Soochow University, Medical Center of Soochow University, Suzhou 215123, China; 18252107296@163.com (X.G.); zwg7807@suda.edu.cn (W.Z.); 4School of Education, Soochow University, Suzhou 215123, China; chenpeiqi0625@163.com (P.C.); zyuqing1021@163.com (Y.Z.); rxy13271569840@163.com (X.R.)

**Keywords:** autistic children, maternal average energy rank of normalized amplitudes of frequency fluctuations, maternal energy rank variability of normalized amplitudes of frequency fluctuations, progressive self–other transposition group intervention, transposition ability

## Abstract

The self-to-other model of empathy (SOME) states that a key reason for the empathic deficiency in autistic individuals is the imbalance of the self–other switch. The existing interventions of theory of mind contain training of self–other transposition ability but combined with other cognitive trainings. The self–other distinction brain areas of autistic individuals have been revealed, but the brain areas of the self–other transposition ability and its intervention have not been investigated. There are normalized amplitudes of low-frequency fluctuations (mALFFs) within 0.01–0.1 Hz and many normalized amplitudes of frequency fluctuations (mAFFs) within 0–0.01, 0.01–0.05, 0.05–0.1, 0.1–0.15, 0.15–0.2, and 0.2–0.25 Hz. Therefore, the current study established a progressive self–other transposition group intervention to specifically and systematically improve autistic children’s self–other transposition abilities. The transposition test with a three mountains test, an unexpected location test, and a deception test was used to directly measure autistic children’s transposition abilities. The Interpersonal Responsiveness Index Empathy Questionnaire with perspective-taking and fantasy subscales (IRI-T) was used to indirectly measure autistic children’s transposition abilities. The Autism Treatment Evaluation Checklist (ATEC) was used to measure autistic children’s autism symptoms. The experiment was designed with two (intervention: experimental group vs. control group) independent variables and two (test time: pretest vs. posttest or tracking test) × three (test: transposition test vs. IRI-T test vs. ATEC test) dependent variables. Furthermore, it used eyes-closed resting-state functional magnetic resonance imaging to investigate and compare the relevant maternal mALFFs and average energy rank and energy rank variability of mAFFs of autistic children’s transposition abilities, autism symptoms, and intervention effects. The results showed the following: (1) There were many improvements (pretest vs. posttest or tracking test) greater than chance 0 in the experimental group, such as the three mountains, lie, transposition, PT, IRI-T, PT tracking, cognition, behavior, ATEC, language tracking, cognition tracking, behavior tracking, and ATEC tracking improvements. However, there was no improvement greater than chance 0 in the control group. (2) The maternal mALFFs and maternal average energy rank and energy rank variability of mAFFs could predict the autistic children’s transposition abilities, autism symptoms, and intervention effects with some overlap and some difference in maternal self–other distinction, sensorimotor, visual, facial expression recognition, language, memory and emotion, and self-consciousness networks. These results indicated that the progressive self–other transposition group intervention successfully improved autistic children’s transposition abilities and reduced their autism symptoms; the intervention effects could be applied to daily life and last up to a month. The maternal mALFFs, average energy rank, and energy rank variability of mAFFs were three effective neural indictors of autistic children’s transposition abilities, autism symptoms, and intervention effects, and the average energy rank and energy rank variability of mAFFs were two new neural indictors established in the current study. The maternal neural markers of the progressive self–other transposition group intervention effects for autistic children were found in part.

## 1. Introduction

Autism spectrum disorder (ASD) is a common group of neurodevelopmental disorders (ND) that frequently occur in children. Patients present with impaired social interaction and communication skills as well as repetitive, stereotypical interests, behaviors, or movements [1]. The incidence of ASD among school-age children in the United States was 2% [2]. Prevalence of ASD among children aged 8 years was 1/68 in 2010 [3] and rose to 1/54 in 2016 [4]. The number of autistic individuals (identity first language, IFL) [5,6] in China exceeded 10 million, of which 1.5 million were children [7,8]. Autistic symptoms make it difficult for autistic children to adapt to life and integrate into society, and this brings great pain to themselves and their families [1,9,10,11], hence the great need for intervention [12].

According to the self-to-other model of empathy (SOME) [13], the process of empathy includes five representation systems, namely situational understanding, affective cue classification, theory of mind, affective representation, and mirror neurons; furthermore, they are affected by the self–other switch (see Figure 1). One key reason for the empathic deficiency in autistic individuals is the imbalance of the self–other switch. For ordinary people, the default state of the self–other switch is “self”, and when activated, it transforms from “self” to “others”. It is difficult for autistic individuals to activate this system, leading them to focus too much on themselves and their interests and ignore the emotions and feelings of others. Even when they notice others, it is difficult for them to recognize the difference between themselves and others and revert from “others” back to “self”; therefore, they are sometimes overly affected by the other person’s state and show excessive emotional reactions [14,15]. Structured laboratory studies often give specific tasks so that autistic individuals can be instructed to activate the self–other switch. When their empathic representation systems are relatively intact, they can show a good level of empathy. However, in daily life, complex information and lack of clear instructions affect the function of the self–others switch, leading to a deficiency in empathy in autistic individuals. The SOME can explain the autistic individuals’ empathy characteristics and the differences between laboratory research results and clinical practice; however, it needs to be verified through empirical intervention studies [16].

Regarding brain mechanisms, three areas are considered responsible for the empathy issues related to ASD: the theory of mind (ToM) brain areas are impaired, such as the medial prefrontal lobe, posterior cingulate gyrus, and bilateral temporoparietal joint area [17,18,19]; the social brain areas are impaired, such as the orbital frontal cortex, middle temporal gyrus, amygdala, and fusiform gyrus [20,21,22]; and the mirror neuron system brain areas are impaired, such as the premotor area, primary motor cortex, inferior parietal lobe, inferior frontal gyrus, superior temporal sulcus, medial temporal lobe, insula, and cingulate gyrus [23,24,25]. The self–other distinction brain areas of autistic individuals include the right supramarginal gyrus, right temporo-parietal junction [26,27,28], superior temporal sulcus and gyrus [26,29,30], medial prefrontal cortex, posterior cingulate cortex, precuneus, hippocampus, temporal pole [31,32,33,34,35,36,37], amygdala, insula, inferior parietal lobule, inferior frontal gyrus, somatosensory, putamen, thalami, and sensorimotor cortices [30,38,39,40,41].

Although ASDs are generally assumed to be lifelong, between 3% and 25% of autistic children reportedly achieved the optimal outcome in losing their ASD diagnosis and entering the normal range of cognitive, adaptive, and social skills by normalizing input by forcing attention outward or enriching the environment, promoting the reinforcement value of social stimuli, preventing interfering behaviors, mass practice of weak skills, and reducing stress and stabilizing arousal [42,43]. Early intervention has a positive effect on them [9,44,45,46]. However, only an unclear minority of autistic children eventually achieved optimal outcome, especially children whose diagnosis was already nuanced, without cognitive impairment, and, in general, with high levels of functioning. Even the literature on super-responders does not show such an optimistic picture in the general sense [47]. Of course, every intervention is worth developing as long as it can reduce the autistic children’s autism symptoms or improve their cognitive, social, and life skills, even if it does not produce the optimal outcome.

The intervention of theory of mind (ToM) is a greatly effective intervention [48,49,50,51,52,53,54,55] that includes the following: looking at things from a different perspective, making assumptions about other people’s intentions and feelings and predicting their behavior, identifying basic and complex emotions, distinguishing between what is real and fake, perspective taking and transpositional consideration, pretend and roleplay games, and social stories. A ToM intervention was set up focusing on social competences and the development of ToM [55]. This intervention is divided into stages: (1) learning the ability to listen to others, making friends, developing visual and auditory perception, and verbal and movement imitation; (2) learning how to distinguish fiction from reality and the assessment of the situation; (3) learning to recognize emotions; and (4) learning to assess a problem from different perspectives and developing the ability to recognize the thoughts and feelings of others. The ToM intervention can obtain an improvement in tests assessing the theory of mind [48]. However, it takes too long, with a duration of 16 weeks. Therefore, a Mini ToM intervention was developed with 8-week and 1-hour-a-week social–cognitive training, including understanding differences between people, reading intentions to predict the behavior of others, recognizing simple and complex emotions, and taking the perspective of other people [53]. The Mini ToM intervention also can improve the ability of ToM [53,56], but it could not improve other autism symptoms. As shown above, the interventions of ToM contained interventions of self–other transposition ability but combined with other cognitive interventions, which might dilute the intervention effect of self–other transposition ability. Therefore, the progressive self–other transposition group intervention should be designed specifically and systematically for the self–other transposition ability according to the key factor—the self–other switch of SOME. Six factors should be given attention in the progressive self–other transposition group intervention for autistic children: (1) Common attention deficit is an important cause of social and language impairment. The common attention deficit of autistic children is early and common, and the common attention deficit of gesture hinders their language development. Gesture guidance can improve the common attention of autistic children in intervention [57]. (2) Autistic children have narrow interests. Integrating game elements into the progressive self–other transposition group intervention is more likely to attract children’s attention and improve their participation [58,59]. (3) Group activities can promote social development. Parents, coaches, researchers, volunteers, and peers are required to participate and cooperate [60,61,62] because parent and teacher scaffolding is paramount for empathy development when children transition from highly scaffolded interactions with their parents and towards interactions with their peers in toddlerhood [63]. Parents’ participation also can help autistic children deal with out-of-control emotions during the intervention tasks through some embodied and verbal approaches for which other persons were not fit, such as a tight hug [64,65]. (4) The difficulty, richness, and interaction of the progressive self–other transposition group intervention should be gradually progressive or step by step [62,66]. (5) The frequency and intensity of the intervention need to be large enough to produce a learning effect. (6) The transposition ability, including cognitive transposition, emotional transposition, and role transposition, should be comprehensively intervened upon [67,68].

The self–other distinction brain areas of autistic individuals have been revealed, but the brain areas of the self–other transposition ability and its intervention have not been investigated. On one hand, the self–other transposition ability contains not only the self–other distinction but also the transposition between self and other. On the other hand, the brain areas of the self–other transposition ability and its intervention effects may not exactly overlap. Finding the neural markers of intervention effects is of great value to reveal the brain mechanism of intervention and develop intervention methods targeting brain function in the future. Furthermore, maternal brain activity may influence autistic children’s self–other transposition abilities, autism symptoms, and intervention outcomes, but it has not been examined. There are many normalized amplitudes of frequency fluctuations (mAFFs) in the ranges of 0–0.01, 0.01–0.05, 0.05–0.1, 0.1–0.15, 0.15–0.2, and 0.2–0.25 Hz in the resting state. The signals of 0.01–0.2 Hz were divided into four frequency bands of 0.01–0.05, 0.05–0.1, 0.1–0.15, and 0.15–0.2 Hz, and the energy of 0.01–0.05 Hz signals was the highest and 0.15–0.2 Hz signals was the lowest [69]. The fitting of the energy curve of the signals in the whole frequency band obeyed a 1/f distribution. The 0.01–0.05 Hz signals were primarily distributed in the prefrontal, parietal, and occipital lobes; 0.05–0.1 Hz signals in the thalamus and basal ganglia; 0.1–0.15 Hz signals in the insula and temporal lobe; and 0.15–0.2 Hz signals in the insula, temporal lobe, and subcortical areas. The 0–0.01 and 0.2–0.25 Hz signals were also associated with some brain areas and cognitive functions [70,71,72]. While researchers found that multiple-frequency bands in different brain areas interacted with each other to affect the same cognitive function [70,72,73,74,75,76,77], it was not quantified how, in some cases, different frequency bands could affect the same cognitive function in some common brain areas through division, cooperation, interaction, and integration. As mAFF represents the ratio of a voxel AFF (amplitudes of frequency fluctuation) value to the average brain AFF value to measure its energy rank, for the first time, we proposed the mean *M* of mAFFs among multiple-frequency bands to quantify the average energy rank of an anatomical automatic labeling (AAL) [78] brain area among multiple-frequency bands and the standard deviation *SD* of mAFFs among multiple-frequency bands to quantify the energy rank variability of an AAL brain area among multiple-frequency bands, which should yield the degree of energy rank homogeneity (diversity) to reveal the division, cooperation, interaction, and integration of multiple-frequency bands.

Therefore, the current study drew on the ToM intervention and the Mini ToM intervention to establish the progressive self–other transposition group intervention to specifically train and improve the self–other transposition abilities [13,16] for autistic children by meeting the requirements of the above six factors. The intervention was hypothesized to improve autistic children’s self–other transposition ability and reduce autism symptoms. Furthermore, the current study hypothesized the maternal mALFFs, average energy rank (*M*), and energy rank variability (*SD*) of mAFFs among multiple-frequency bands in the same AAL brain areas in a resting state could predict autistic children’s self–other transposition ability, autism symptom, and progressive self–other transposition group intervention effect.

## 2. Materials and Methods

### 2.1. Participants

Twenty-seven autistic children diagnosed with ASD by the hospital and with 36–95 scores in the Autism Treatment Evaluation Checklist (ATEC), with age 57–153 months, participated in the experiments. Sixteen autistic children and their mothers participated in an experimental group with progressive self–other transposition group intervention. There were 2 girls and 14 boys with age *M* ± *SD* = 100.5 ± 27.7 months, Combined Raven’s Test (CRT) *M* ± *SD* = 97.75 ± 18.85, and ATEC *M* ± *SD* = 66.5 ± 15.87 and 16 mothers with age *M* ± *SD* = 37.31 ± 4.98 years. Only the mothers participated in functional magnetic resonance imaging (fMRI) scanning. All the mothers met the criteria for fMRI scanning: They had no metal implants, were not claustrophobic, and had a head size compatible with the head coil. Eleven autistic children took part in a control group, including two girls and nine boys, with age *M* ± *SD* = 90.91 ± 21.03 months, CRT *M* ± *SD* = 85.18 ± 18.67, and ATEC *M* ± *SD* = 60.09 ± 12.55. There was no significant difference in age, CRT, or ATEC between the experimental and control groups (*p*s > 0.05). All the autistic children participated with their mothers’ permission. The mothers volunteered to participate, each completing an informed consent form before the experiments. The current study was approved by the Research Organisms Life Safety and Body Use Ethics Committee, School of Education in Soochow University, China, and was in accordance with the ethical guidelines of the Declaration of Helsinki revised version in 2013.

### 2.2. Experimental Procedures

In experimental group, the mothers participated in structural imaging and eyes-closed resting-state fMRI scanning before the intervention. The progressive self–other transposition group intervention for autistic children (see Table 1) lasted six weeks and was carried out thrice per week; each intervention lasted 1 h, resulting in a total of 18 h. Each intervention included warm-up games for 5 min, cognitive transposition, emotional transposition and role transposition games for 15 min each, and concluding games for 10 min. The cognitive, emotional, and role transposition games followed the Latin square order and changed each time. Each game was guaranteed to be ranked first, second, and third for six times. The interventions were divided into three stages, including self-centeredness and embody transposition in the primary stage (1–6 times), embodied transposition and other-centeredness in the middle stage (7–14 times), and other-centeredness and other-expectation in the high stage (15–18 times). The transposition cognition complexity of the three stages gradually increased to ensure the improvement of autistic children’s transposition cognition abilities step by step. Each autistic child was assigned two assistants including his/her mother and one master or undergraduate student in psychology. The same exercise was first demonstrated by the coach and then by the assistants to help the autistic children established common attention through multi-person demonstration. The autistic children imitated games with the help of assistants, who guided their attention, assisted with the completion of steps, prevented children from running around, and helped autistic children deal with out-of-control emotions during the intervention tasks. The intervention expert team provided systematic training to the coach and assistants before the intervention; observed, guided, and corrected their language and behavior throughout the intervention period; and observed the performance of autistic children to find and solve the existing problems, thus forming new intervention programs.

In the control group, autistic children did not receive progressive self–other transposition group intervention; however, their parents took them to other interventions in other institutions. After the intervention on the experimental group, the control group was treated as the new experimental group for other interventions by our team.

### 2.3. Tests

The transposition test included a three mountains task measuring the embodied self-other translation ability with a test-retest reliability of 0.78 [79,80], an unexpected location transfer task measuring the cognitive self–other translation ability with the a test-retest reliability of 0.76 [81,82,83], and a deception task measuring the operational self–other translation ability [84,85,86], which added up to form an index of transposition ability. The three mountains task required autistic children to point out what they saw from their perspectives and the perspectives of others. They need to transform the space between their embodied perspectives and others’ embodied perspectives. The unexpected location transfer task required autistic children to point out where they actually knew things and where others remembered them. They need to transform memory between their perspectives and others’ perspectives. The deception task required autistic children to deceive the robber by telling him the wrong place of things to prevent the puppet’s things from being robbed and show the king the correct place of things to let him give the puppet the reward. They need to transform intention between their perspectives and perspectives of the king, the robber, and the puppet to operate their words and actions.

In the Interpersonal Responsiveness Index Empathy Questionnaire (IRI) [87,88], the perspective-taking (PT) subscale measures an individual’s spontaneous tendency to adopt the ideas of others, and the fantasy (FS) subscale measures an individual’s response to empathy with a fictional character. They were added up to form the IRI transposition ability (IRI-T). There are seven items both in the PT and FS subscales. Using a 5-point scoring method, the PT and FS internal consistency coefficients were 0.75 and 0.78, respectively, and the PT and FS test-retest reliabilities were noted as 0.61 (male) and 0.62 (female) and 0.79 (male) and 0.81 (female), respectively. The exploratory factor analysis found that the construct validity of the scale was good: χ2/df = 2.33, CFI = 0.90, IFI = 0.90, RMSEA = 0.054.

The Autism Treatment Evaluation Checklist (ATEC) [89,90] is used to assess autism symptoms and their changes and the efficacy of treatment for children aged 2–12 years. It consists of 77 items and four subscales: speech/language/communication (henceforth, language); social interaction; feeling/cognition/consciousness (henceforth, cognition); and health/body/behavior (henceforth, behavior). The first three subscale options range from 0 (none) to 2 (often), and the fourth subscale options range from 0 (none) to 3 (heavy). The higher the score, the more severe the autism symptom. The score ranges from 20 to 49 for mild, 50–79 for moderate, and more than 80 for severe. The internal consistency coefficient of the ATEC scale is 0.93. ATEC scores remained relatively stable and were correlated with *r* = 0.80 between the first and subsequent assessments (5.6 years old and 5–6 years later) [91].

In the experimental group, the autistic children took the transposition test administered by a researcher before (pretest) and after (posttest) the six-week intervention to directly test their transposition abilities and improvements. The autistic children’s mothers filled out the IRI-T test and the ATEC test for autistic children before, after, and a month after (tracking test) the intervention to indirectly test the autistic children’s transposition abilities, autism symptoms, and their improvements. In the control group, the autistic children’s mothers filled out the IRI-T test and the ATEC test for autistic children before and after 6 weeks to indirectly test the autistic children’s transposition abilities, autism symptoms, and their improvements.

The experiment adopted a 2 (intervention: experimental group vs. control group) × 2 (test time: pretest vs. posttest or tracking test) × 3 (test: transposition test vs. IRI-T test vs. ATEC test) hybrid design. The first one was the independent variable, and the last two were dependent variables. The experiment was designed to investigate whether and how the intervention changed scores on the three tests. 

### 2.4. Behavioral Data Analysis

The behavioral data analysis contained three steps: (1) The first was descriptive statistics—calculating the mean and standard deviation of the pretest, improvement (pretest vs. posttest), and tracking improvement (pretest vs. tracking test) of the transposition test, the IRI-T test, and the ATEC test in experimental group as well as the pretest and improvement of the IRI-T and ATEC tests in control group. The improvements were used to quantitatively measure the difference comparing posttest or tracking test to pretest, such as posttest or tracking test minus pretest for the transposition test and IRI-T test and pretest minus posttest or tracking test for the ATEC test. (2) The second was the analysis of variance and single-sample *t*-test—analyses of variance were performed to find whether there was any difference between the IRI-T test or ATEC test or their improvements between experimental group and control group. Repeated-measures analyses of variance were carried out for the IRI-T and ATEC tests (pretest vs. posttest) in the experimental group or control group to find whether there was any improvement. Then, the improvements and tracking improvements of the transposition, IRI-T, and ATEC tests in the experimental group were compared with chance 0 to show whether the progressive self–other transposition group intervention could change them. Furthermore, the improvements of the IRI-T and ATEC tests in the control group were compared with chance 0 to determine whether other interventions in different institutions, daily life, growth, and other interference factors could change them. The qualitative difference between the experimental and control groups was compared to examine whether the intervention effect in experimental group really existed, that is, whether the improvements were greater than 0 only in the experimental group. (3) Third was the correlation analysis—the Pearson correlation between maternal brain activity with the pretest and improvement of the transposition, IRI-T, and ATEC tests of autistic children in the experimental group was investigated to reveal the influence of maternal brain activity on autistic children’s intervention. Pretest and improvement information already included posttest information, so posttest correlation analysis was not performed.

### 2.5. Resting-State Data Collection and Analysis

The fMRI data were collected using a GE SIGNA™ Architect 3.0 T magnetic resonance imaging scanner (General Electric Company, American) and an 8-channel phased front head coil. The structural imaging used a 3D T1 BRAVO sequence with sagittal scans. The scanning parameters were as follows: the phase encoding direction was A/P, TR = 7.7 ms, TE = 3.1 ms, FA = 12°, FOV = 256 × 256 mm^2^, total 100 layers. The eyes-closed resting-state imaging used gradient echo (GRE) single-excitation echo-planar imaging (EPI), with the following scan parameters: the phase encoding direction was R/L, TR = 2000 ms, TE = 30 ms, FA = 90°, FOV = 220 × 220 mm^2^, matrix size = 64 × 64, depth = 3 mm, planar resolution = 3.44 × 3.44 mm^2^, interval scanning, 33 layers, layer spacing = 0.6 mm, total 240 layers. The inaccuracy and tolerance of the experimental equipment used in this inquiry were small enough for the scans of the brain, and they were corrected by pretreatment and analysis. All the participants first received the structural scan followed by the eyes-closed resting-state scan.

Pretreatment and analysis of resting-state data used DPARSF 3.0 Advanced Edition Calculate [92] in Original Space (Warp by DARTEL), following standard procedures: (1) First was the conversion of raw DICOM-format data to NIFTI format. To allow for signal stabilization of the image, the first 10 TR images were removed, after which time layer correction (slice timing) and head movement correction (realignment, adopting Friston 24) were applied. If a head movement greater than 3 mm or 3 degrees occurred during the resting state, the data were deleted. (2) The new segment + DARTEL was used to split the structural T1 data without standardization and register the T1 split data directly to the resting-state functional images. Before the registration of the structural and functional data, the AC-PC line of each participant’s T1 image and resting-state function was registered, and then automatic registration was applied. Therefore, the resting-state analysis took place in the original T1 space. (3) A regression analysis was conducted, adjusting for head motion, linear drift, white matter, and cerebrospinal fluid. (4) The normalized amplitude of frequency fluctuation (mALFFs, filter bands: 0.01–0.1Hz; mAFFs, filter bands: 0–0.01, 0.01–0.05, 0.05–0.1, 0.1–0.15, 0.15–0.2, and 0.2–0.25 Hz) was calculated. (5) The resting-state function was registered to the standard MNI space (normalized by DARTEL), using Bounding Box (−90 −126 −72; 90 90 108) and a 3 × 3 × 3 mm^3^ voxel size, with 4 × 4 × 4 mm^3^ full width at half maximum (FWHM) smoothing.

REST1.8 [93] was first used to extract the mALFFs in 0.01–0.1Hz and the mAFFs in the six frequency bands in 116 AAL [78] brain areas. The *M* and *SD* of mAFFs in the six frequency bands were calculated in each AAL brain area. A Pearson correlation analysis was conducted between the mALFFs and the *M* and *SD* of mAFFs with the progressive self–other transposition group intervention effect with 3 (mALFFs vs. *M* of mAFFs vs. *SD* of mAFFs) × 3 (transposition vs. IRI-T vs. ATEC) × 2 (pretest vs. improvement) = 18 variable levels. The correlation analysis was used to investigate whether and how maternal brain activity predicted autistic children’s transposition ability, IRI-T ability, ATEC symptoms, and their improvements (the progressive self–other transposition group intervention effect). Since the original mALFF and mAFF for each AAL brain area (the average mALFF or mAFF of its all voxels) were extracted [94,95,96]), a multiple-comparisons correction was unnecessary and could not be made for the correlation analyses above [97,98]. The overlap and difference between the *M* or *SD* of mAFFs and mALFFs in relevant brain areas of the intervention effect were calculated. The relevant brain areas were visualized with the BrainNet Viewer (http://www.nitrc.org/projects/bnv/; accessed on 31 October 2019) [99]. 

## 3. Results

There were 16 valid autistic children in the progressive self–other transposition group intervention, 16 valid mothers in the eyes-closed resting state, and 11 valid autistic children in the control group.

### 3.1. Behavioral Results

The indexes in the experimental and control groups as well as their comparison are shown in Table 2. The indexes of the IR-T and ATEC pretests did not differ significantly between the two groups in the independent-sample *t*-test except that the behavior of the ATEC in experimental group was greater than that in control group. 

A 2 (intervention: experimental group vs. control group) × 2 (IRI-T test: pretest vs. posttest) analysis of variance was performed to find that there was neither a significant group main effect nor interaction effect, *p*s > 0.05, but there was a significant test main effect, *F* (1, 26) = 7.82, *p* < 0.01, η_p_^2^ = 0.238, and IRI-T posttest was significantly greater than IRI-T pretest. A 2 (intervention: experimental group vs. control group) × 2 (ATEC test: pretest vs. posttest) analysis of variance was performed to find that there was neither a significant main effect nor interaction effect, *p*s > 0.05. Two repeated-measures analyses of variance were carried out for the IRI-T and ATEC tests (pretest vs. posttest) in the experimental group. Sphericities were significant (*p* < 0.001); then, greenhouse corrections were made, and there was a significant main effect of each test: for the IRI-T test, *F* (1, 15) = 8.55, *p* < 0.01, η_p_^2^ = 0.363, and the posttest was significantly greater than the pretest; for the ATEC test, *F* (1, 15) = 4.78, *p* < 0.05, η_p_^2^ = 0.242, and posttest was significantly less than the pretest. Two repeated-measures analyses of variance were carried out for the IRI-T and ATEC tests (pretest vs. posttest) in the control group to find that there was no significant main effect, *p*s > 0.05, and each posttest was not different with the pretest. 

The improvements were used to quantitatively measure the difference comparing the posttest or tracking test to the pretest. The single-sample *t*-test for the improvements had the same results as the repeated-measures analysis of variance between the posttest or tracking test with the pretest, but the single-sample *t*-test was simpler, more intuitive, and more quantifiable. There were many improvements greater than chance 0 in the experimental group, as shown by the single-sample *t*-test in Table 2, such as the three mountains, lie, transposition, PT, IRI-T, PT tracking, cognition, behavior, ATEC, language tracking, cognition tracking, behavior tracking, and ATEC tracking improvements. However, there was no improvement greater than chance 0 in the control group. The ATEC scores in the posttest and tracking test in the experimental group were 27–84 and 37–85, none of which was less than 20 in order to be normal. The ATEC scores in the posttest in the control group were 29–92, none of which was less than 20 in order to be normal as well.

Pearson correlation was performed for the transposition, IRI-T, and ATEC pretests in the experimental group. There was no significant correlation among them (*p*s > 0.05), indicating no multicollinearity. The lie pretest was positively related to the PT pretest, *r* = 0.51, *p* < 0.05, indicating that the transposition ability and IRI-T ability had a correlation but more differences, and they could not be added up. Therefore, it was necessary to separately examine the relevant brain areas of the transposition, IRI-T, and ATEC pretests and their improvements. 

### 3.2. Resting-State Brain Activity

#### 3.2.1. mALFFs Related to the Pretest and Improvement

There were many brain areas whose mALFFs were related to the pretest (see Table 3 and Figure 2). Transposition ability in the pretest was positively related to some brain areas of the self–other distinction network (Frontal_Sup_L, Frontal_Mid_L, Frontal_Mid_R, Frontal_Inf_Oper_L, Frontal_Inf_Tri_L, and Frontal_Sup_Medial_R) [26,27,28,35,38,40,41] but was negatively related to some brain areas of the sensorimotor (Cerebelum_6_L and Cerebelum_6_R) [97,100] and visual and facial expression recognition networks (Calcarine_L, Calcarine_R, Lingual_L, Lingual_R, and Fusiform_R) [98,101,102,103]. IRI-T ability in the pretest was positively related to some brain areas of the memory and emotion (ParaHippocampal_R) [98] and sensorimotor (Vermis_10) networks but was negatively related to some brain areas of self–other distinction (Parietal_Inf_L, Parietal_Inf_R, and SupraMarginal_R) and language (Lingual_L, Lingual_R, Parietal_Inf_L, Parietal_Inf_R, SupraMarginal_R, and Angular_L) [104] networks. ATEC symptom in the pretest was positively correlated to some brain areas of self–other distinction (Parietal_Inf_L, Parietal_Inf_R, and SupraMarginal_R), language (Parietal_Inf_L, Parietal_Inf_R, and SupraMarginal_R), self-consciousness (Precuneus_L) [97,105,106], and sensorimotor (Putamen_L, Putamen_R, Pallidum_L, and Cerebelum_8_R) networks.

There were many brain areas related to improvement (see Table 3 and Figure 2): Transposition ability improvement was positively related to some brain areas of the visual and facial expression recognition (Lingual_L, Lingual_R, and Fusiform_R) and sensorimotor (Cerebelum_6_L and Cerebelum_6_R) networks but was negatively related to some brain areas of the self–other distinction network (Frontal_Sup_L, Frontal_Inf_Oper_L, and Frontal_Sup_Medial_R). IRI-T ability improvement was positively related to some brain areas of the memory and emotion (Hippocampus_L and Temporal_Inf_L), face expression recognition (Fusiform_L), and sensorimotor (Cerebelum_6_L and Cerebelum_8_L) networks but was negatively related to some brain areas of the visual (Cuneus_L, Cuneus_R, and Occipital_Sup_R) [101,102,103] and self-consciousness (Precuneus_R) networks. ATEC symptom improvement (reduction) was positively correlated to a brain area of the self–other distinction network (Frontal_Inf_Tri_R) but was negatively correlated to a brain area of sensorimotor network (Cerebelum_Crus2_R).

#### 3.2.2. *M* of mAFFs Related to the Pretest and Improvement

There were many brain areas whose *M*s of mAFFs were related to the pretest (see Table 4 and Figure 3). Transposition ability was positively correlated to some brain areas regarding the self–other distinction network (Frontal_Sup_L, Frontal_Mid_L, Frontal_Inf_Tri_L, and Frontal_Mid_Orb_R); however, it was negatively correlated to some brain areas related to the visual and facial expression recognition (Calcarine_L, Calcarine_R, Lingual_L, Lingual_R, and Fusiform_R) and sensorimotor (Cerebelum_6_L and Cerebelum_6_R) networks. IRI-T ability was positively correlated to some brain areas specific to the self–other distinction (Temporal_Pole_Sup_L), memory and emotion (ParaHippocampal_R and Temporal_Pole_Sup_L), and sensorimotor (Vermis_10) networks; however, it was negatively correlated to some brain areas of the self–other distinction (Parietal_Inf_L and Parietal_Inf_R), visual and language (Lingual_L, Lingual_R, Occipital_Inf_L, Postcentral_L, Parietal_Inf_L, Parietal_Inf_R, and Angular_L), as well as memory and emotion (Temporal_Inf_R) networks. ATEC symptom was positively correlated to some brain areas related to the self–other distinction (Parietal_Inf_L and Parietal_Inf_R), visual and language (Occipital_Inf_L, Parietal_Inf_L, Parietal_Inf_R, and Angular_L), self-consciousness (Precuneus_L), memory and emotion networks (Temporal_Inf_L), and sensorimotor (Putamen_L, Pallidum_L, and Cerebelum_8_R) networks.

There were many brain areas whose *M*s of mAFFs were related to improvement (see Table 4 and Figure 3). Transposition ability improvement was positively correlated to some brain areas of the visual and facial expression recognition (Lingual_L and Fusiform_R) and sensorimotor (Cerebelum_6_R) networks; however, it was negatively correlated to some brain areas related to the self–other distinction network (Frontal_Sup_L, Frontal_Inf_Oper_L, Frontal_Inf_Oper_R, Frontal_Inf_Tri_L, Frontal_Sup_Medial_L, and Frontal_Sup_Medial_R). IRI-T ability improvement was positively correlated to some brain areas of the memory and emotion (Hippocampus_L), visual and facial expression recognition (Fusiform_L), and sensorimotor (Rolandic_Oper_L, Cerebelum_6_L, and Cerebelum_8_L) networks; however, it was negatively correlated to some brain areas related to the visual (Cuneus_L, Cuneus_R, and Occipital_Sup_R) and self-consciousness (Precuneus_R) networks. ATEC symptom improvement was positively correlated to some brain areas related to the self–other distinction (Frontal_Mid_R, Frontal_Mid_Orb_R, and Putamen_R) and memory and emotion (Temporal_Inf_L) networks.

#### 3.2.3. *SD* of mAFFs Related to the Pretest and Improvement

There were many brain areas whose *SD*s of mAFFs were related to the pretest (see Table 5 and Figure 4). Transposition ability was positively correlated to some brain areas related to the self–other distinction (Frontal_Sup_L, Frontal_Sup_R, Frontal_Mid_L, Frontal_Mid_R, Frontal_Mid_Orb_L, Frontal_Mid_Orb_R, Frontal_Inf_Oper_L, Frontal_Inf_Tri_L, Frontal_Sup_Medial_L, and Frontal_Sup_Medial_R) and sensorimotor (Cerebelum_4_5_L) networks; however, it was negatively correlated to some brain areas responsible for the visual (Calcarine_L and Calcarine_R), memory and emotion (Temporal_Mid_L), and sensorimotor (Postcentral_L, Cerebelum_Crus1_L, Cerebelum_Crus2_L, Cerebelum_7b_L, and Cerebelum_8_L) networks. IRI-T ability was positively correlated to some brain areas belonging to the memory and emotion (Heschl_R) and the sensorimotor (Cerebelum_9_L and Vermis_10) networks; however, it was negatively correlated to some brain areas related to the language (Parietal_Sup_L) and sensorimotor (Rolandic_Oper_L) networks. ATEC symptom was positively correlated to some brain areas belonging to the self–other distinction (Temporal_Pole_Mid_R), visual and facial expression recognition (Cuneus_L, Occipital_Sup_L, Fusiform_R, and Angular_L), memory and emotion (Temporal_Pole_Mid_R), and sensorimotor (Vermis_8) networks.

There were many brain areas whose *SD*s of mAFFs were related to the improvement (see Table 5 and Figure 4). Transposition ability improvement was positively correlated to some brain areas belonging to the sensorimotor network (Cerebelum_7b_L) while negatively correlated to some brain areas related to the self–other distinction network (Frontal_Mid_Orb_R and Frontal_Inf_Oper_L). IRI-T ability improvement was positively correlated to some brain areas related to the self–other distinction (Parietal_Inf_L), memory and emotion (Heschl_L and Temporal_Inf_L), and sensorimotor (Rolandic_Oper_R and Cerebelum_Crus1_L) networks; however, it was negatively correlated to some brain areas belonging to the visual (Cuneus_R and Occipital_Sup_R), self-consciousness (Precuneus_R), as well as memory and emotion (Temporal_Inf_R) networks. ATEC symptom improvement was positively correlated to some brain areas related to the self–other distinction (Frontal_Mid_Orb_R and Temporal_Pole_Mid_R) and memory and emotion (Temporal_Pole_Mid_R) networks; however, it was negatively correlated to some brain area belonging to the self-consciousness network (Precuneus_R).

#### 3.2.4. Consistency between the *M* or *SD* of mAFFs and mALFFs in Relevant Brain Areas of the Pretest and Improvement

The maternal *M* or *SD* of mAFFs and mALFFs could predict the autistic children’s pretest and improvement with some overlap and some difference in maternal self–other distinction, sensorimotor, visual, facial expression recognition, language, memory and emotion, and self-consciousness networks.

Regarding consistency between mAFFs’ *M* and mALFFs in the pretest (see (a) in Table 6), we found the following: (1) Ten common relevant brain areas of transposition ability were identified in the *M* of both mAFFs and mALFFs. There was one relevant brain area only in the *M* of mAFFs and three relevant brain areas only in mALFFs. (2) Seven common relevant brain areas of IRI-T ability were identified in the *M* of both mAFFs and mALFFs. There were four relevant brain areas only in the *M* of mAFFs and one relevant brain area only in mALFFs. (3) Seven common relevant brain areas of ATEC symptom were identified in the *M* of both mAFFs and mALFFs. There were two relevant brain areas only in the *M* of mAFFs and two relevant brain areas only in mALFFs.

Regarding consistency between mAFFs’ *M* and mALFFs in improvement (see (b) in Table 6), we found the following: (1) Six common relevant brain areas of transposition ability improvement were identified in the *M* of both mAFFs and mALFFs. There were three relevant brain areas only in the *M* of mAFFs and two relevant brain areas only in mALFFs. (2) Eight common relevant brain areas of IRI-T ability improvement were identified in the *M* of both mAFFs and mALFFs. There was one relevant brain area only in the *M* of mAFFs and one relevant brain area only in mALFFs. (3) No common relevant brain areas of ATEC symptom improvement were identified in the *M* of both mAFFs and mALFFs. There were four relevant brain areas only in the *M* of mAFFs and two relevant brain areas only in mALFFs.

Regarding consistency between mAFFs’ *SD* and mALFFs in pretest (see (a) in Table 7), we found the following: (1) Eight common relevant brain areas of transposition ability were identified in the *SD* of both mAFFs and mALFFs. There were 11 relevant brain areas only in the *SD* of mAFFs and 5 relevant brain areas only in mALFFs. (2) One common relevant brain area of IRI-T ability was identified in the *SD* of both mAFFs and mALFFs. There were four relevant brain areas only in the *SD* of mAFFs and seven relevant brain areas only in mALFFs. (3) One common relevant brain area of ATEC symptom was identified in the *SD* of both mAFFs and mALFFs. There were five relevant brain areas only in the *M* of mAFFs and eight relevant brain areas only in mALFFs.

Regarding consistency between mAFFs’ *SD* and mALFFs in improvement (see (b) in Table 7), we found the following: (1) One common relevant brain area of transposition ability improvement was identified in the *SD* of both mAFFs and mALFFs. There were two relevant brain areas only in the *SD* of mAFFs and seven relevant brain areas only in mALFFs. (2) Four common relevant brain areas of IRI-T ability improvement were identified in the *SD* of both mAFFs and mALFFs. There were five relevant brain areas only in the *SD* of mAFFs and five relevant brain areas only in mALFFs. (3) No common relevant brain areas of ATEC symptom improvement were identified in the *SD* of both mAFFs and mALFFs. There were three relevant brain areas only in the *SD* of mAFFs and two relevant brain areas only in mALFFs.

## 4. Discussion

### 4.1. The Progressive Self–Other Transposition Group Intervention Effect

In the experimental group, the three mountains, lie, transposition, PT, IRI-T, PT tracking, cognition, behavior, ATEC, language tracking, cognition tracking, behavior tracking, and ATEC tracking improvements were greater than chance 0, which indicates that the progressive self–other transposition group intervention successfully improved autistic children’s transposition abilities in both direct and indirect measurements as well as rectified their autism symptoms, while the intervention effects could last at least one month after the intervention. However, there was no improvement of IRI-T or ATEC greater than chance 0 in the control group, which indicates that interventions in other institutions, daily life, growth, and other interference factors could not improve autistic children’s transposition abilities or rectify their autism symptoms; thus, the progressive self–other transposition group intervention effect in the experimental group did exist, including improvements of transposition ability, IRI-T ability, and ATEC symptoms.

The transposition test measures the transposition ability in the laboratory; however, the IRI-T measures the transposition ability in daily life, while the ATEC measures the autism symptom in daily life. Therefore, the progressive self–other transposition group intervention effects existed not only in the laboratory but also in daily life and could last at least one month after the intervention, and its impact existed not only on the ability for improvement of ToM [53,56] but also on other autism symptom improvements, which might be due to several reasons: First, the autistic children’s abilities were deeply improved and able to be transferred and generalized to daily learning and life, which may be because the current study captured the key factor—the self–other switch of SOME [13,16] from the ToM Intervention [55] and the Mini ToM Intervention [56]—to design the intervention specifically and systematically for self–other transposition ability and avoid diluting the effect of the key factor by other factors as in the ToM and Mini ToM interventions. It may also be due to the six factors of the intervention in the current study, namely common attention guidance, game form, group activities, step-by-step progress, enough intervention frequency, and intensity and the three content dimensions. The common attention guidance ensured that autistic children noticed the intervention contents and requirements [57]. The game form ensured that autistic children were interested in participating in the intervention [58,59]. The group activities ensured that the intervention took place in a group social context, which was comprised of simulations of real social life [61,62]. Maternal participation also helped autistic children deal with anxiety, anger, depression, and other out-of-control emotions during the intervention tasks through some embodied and verbal approaches for which other persons were not fit, such as a tight hug [64,65]. The step-by-step progress [62,66], such as self-centeredness, embody transposition, other-centeredness, and other-expectation [67,68], ensured that the transposition abilities were broken down into parts, from easy to difficult, to help autistic children gradually understand and learn its cognitive structure. The three-time-a-week intervention frequency and increasing intensity were enough to produce not only the ability of ToM but also other autism symptom improvements in the current study. The 1-hour-a-week Mini ToM intervention might have enough frequency and intensity to produce only the ability for ToM improvement but no other autism symptom improvement [56]. The three content dimensions, namely cognitive, emotional, and role transpositions, ensured that the transposition abilities were comprehensively intervened upon. Second, the autistic children and their mothers acquired some transposition intervention awareness and techniques to use in their daily learning and life, which may be due to the six factors, especially the involvement of the mothers [66,68]. Third, the autistic children and their mothers acquired some transposition intervention awareness and techniques to learn and explore new transposition and other interventions in their daily learning and life. These reasons warrant further study to promote the maintenance and migration of intervention effects.

### 4.2. mALFF-Relevant Maternal Brain Areas of the Intervention

Regarding the transposition test, the autistic children’s transposition ability [80,83,84] was positively related to their mothers’ self–other distinction [26,27,28,35,38,40,41] and sensorimotor [97,100] networks. If the mothers had high activity of these brain areas in the resting state, they might be good at the self–other distinction and sensorimotor functions, promoting the transposition ability in autistic children. The autistic children’s transposition ability was negatively related to their mothers’ visual and facial expression recognition [101,102,103], olfactory, emotion [98], sensorimotor, and language [104] networks. If the mothers had high activity of these brain areas in the resting state, they might be good at these functions, promoting those in autistic children. Therefore, the autistic children might like to use these functions to accomplish tasks, thus hindering the development of the transposition ability. The maternal self–other distinction and sensorimotor functions were more positively impactful than seeing facial expressions and speaking. In addition, the autistic children’s transposition ability improvement was negatively related to their mothers’ self–other distinction and sensorimotor networks but was positively related to their mothers’ visual and facial expression recognition and sensorimotor networks. The positively/negatively relevant brain areas of the transposition ability were usually the negatively/positively relevant brain areas of the transposition ability improvement. This might come from a baseline effect: the progressive self–other transposition group intervention in the current study had good effect and was particularly friendly to children with severe autism. Therefore, the lower transposition abilities of autistic children could be substantially improved if their mothers have lower activity in the self–other distinction network but higher activity in the visual and facial expression recognition and sensorimotor networks. However, the autistic children with higher transposition abilities could only show improvement in smaller amounts, as their mothers have higher activity in the self–other distinction network but lower activity in the visual and facial expression recognition and sensorimotor networks.

Regarding the IRI-T test, the autistic children’s IRI-T ability [87,88] was positively related to their mothers’ memory and emotion [98] and sensorimotor networks but was negatively related to their mothers’ self–other distinction and language networks. There was a compensating effect that if the autistic children’s IRI-T ability was low, their mother might want to use higher self–other distinction to fix it. If the mother was good at emotion, she might promote her autistic child’s IRI-T ability; but if the mother was good at speaking, she might block her autistic child’s IRI-T ability. The feeling and expression abilities were more positively impactful than speaking abilities. In addition, the autistic children’s IRI-T ability improvement was positively related to their mothers’ memory and emotion, face expression recognition [98], and sensorimotor networks but was negatively related to their mothers’ visual and self-consciousness (self-center) networks. The feeling and expression abilities were more positively impactful than seeing and self-centered abilities.

Regarding the ATEC test, the autistic children’s ATEC symptom [89,90] was positively correlated to their mothers’ self–other distinction, language, self-consciousness, and sensorimotor networks. The maternal speaking, self-centered, and sensorimotor abilities might increase the autistic children’s ATEC symptoms. There was a compensating effect that if the autistic children’s ATEC symptom was high, their mother might want to use higher self–other distinction to fix it. In addition, the autistic children’s ATEC symptom improvement (reduction) was positively correlated to their mothers’ self–other distinction network but was negatively correlated to their mothers’ sensorimotor network. The maternal self–other distinction was more impactful than sensorimotor abilities in reducing the autistic children’s ATEC symptoms. The autistic children’s ATEC symptom and its improvement had different relevant maternal brain networks.

Because both the transposition test and IRI-T test measure the self–other translation ability, according to logic and the existing theory, their relevant maternal brain networks are the relevant maternal brain networks of the self–other transposition ability [13,16]. The self–other transposition ability and its improvement contained not only self–other distinction function [28,40] but also other functions such as sensorimotor, visual, facial expression recognition, language, memory and emotion, and self-consciousness, indicating the necessity of the current study to investigate the relevant brain areas of the self–other transposition ability and its improvement for the first time.

The relevant maternal brain networks of the ATEC symptom and its improvement were similar to those of the self–other translation ability and its improvement, indicating that the self–other translation ability might be the key factor in autism symptoms.

### 4.3. Relevant Maternal Average Energy Rank and Energy Rank Variability of the Intervention

There are some rhythms in each AAL brain area in the resting state, such as mAFFs in 0–0.01, 0.01–0.05, 0.05–0.1, 0.1–0.15, 0.15–0.2, or 0.2–0.25 Hz [69,70,71,72,73,74,75,76,77]. The current study proposed the *M* of the mAFFs as maternal average energy rank and the *SD* of the mAFFs as maternal energy rank variability of each AAL brain area among those six multiple-frequency bands and detected their relationship with autistic children’s progressive self–other transposition group intervention. On the whole, with the maternal average energy rank and energy rank variability of the mAFFs as two new neural indictors, the maternal self–other distinction, facial expression recognition, memory and emotion, self-consciousness, language, visual, and sensorimotor networks could predict the autistic children’s transposition ability, IRI-T ability, and ATEC symptom and their improvements. The self–other distinction was more impactful than self-centeredness, and feeling and expression abilities were more impactful than seeing and speaking.

The average energy rank among multiple-frequency bands could predict many cognitive functions, including the transposition ability, IRI-T ability, ATEC symptom, and the progressive self–other transposition group intervention effect in some AAL brain areas. Some mechanisms may account for this. A relevant AAL brain area was important for a cognitive function, and it was sufficiently strong to draw a substantial amount of energy from other AAL brain areas to produce a high cognitive function, resulting in its positive relevance. In addition, a relevant AAL brain area was necessary for the cognitive function. For some participants, it robbed resources from other necessary brain areas, which worked properly but inefficiently. Therefore, its high energy rank limited the function of other necessary AAL brain areas, producing a low cognitive function. For other participants, working properly with low energy rank was efficient, and it collaborated well with other necessary AAL brain areas. Therefore, a low energy rank could produce a high cognitive function, resulting in negatively relevant AAL brain areas. Furthermore, positively relevant brain areas would become negatively relevant if participants trained their brains to increase their efficiency, therefore requiring less energy. The negatively relevant brain areas would become positively relevant if participants trained their brains to increase their efficiency to prevent them from limiting the functions of other necessary brain areas. The underlying mechanisms are complex and require further research.

The energy rank variability among multiple-frequency bands could predict many cognitive functions, including the transposition ability, IRI-T ability, ATEC symptom, and the progressive self–other transposition group intervention effect in some AAL brain areas. Several mechanisms may account for this. First, in a relevant AAL brain area, six frequency bands required different energy ranks to undertake different cognitive processes, resulting in a high energy rank variability. They subsequently cooperated to produce a high cognitive function. This distributed task system resulted in a positively relevant AAL brain area. Second, in a relevant AAL brain area, the six frequency bands required a similar energy rank to synchronously undertake the same cognitive processes, resulting in low energy rank variability, and produced a high cognitive function. This integrated task system resulted in a negatively relevant AAL brain area.

Previous researchers have mostly paid attention to the different brain areas in which different frequency bands affect the same cognitive function [70,71,72,73,74,75,76,77]; however, they neither attached great importance to nor quantified any potential common brain areas. The current study successfully quantified and confirmed that the average energy rank and energy rank variability among multiple-frequency bands in some common AAL brain areas are associated with the progressive self–other transposition group intervention.

### 4.4. Consistency between the Average Energy Rank or Energy Rank Variability of mAFFs and mALFFs

The maternal *M* of mAFFs and mALFFs had much overlap but little difference in the relevant brain areas of autistic children’s progressive self–other transposition group intervention, which indicates that, using the mALFFs as a calibration, the average energy rank among multiple-frequency bands had good calibration validity (very similar to the calibration) and acceptable ecological validity (some difference with the calibration) as a new indictor in predicting autistic children’s progressive self–other transposition group intervention.

The maternal *SD* of mAFFs and mALFFs had little overlap but much difference in the relevant brain areas of autistic children’s progressive self–other transposition group intervention, which indicates that, using the mALFFs as a calibration, the energy rank variability among multiple-frequency bands had acceptable calibration validity (somewhat similar to the calibration) and good ecological validity (very different from the calibration) as a new indicator in predicting autistic children’s progressive self–other transposition group intervention.

### 4.5. Limitation

None of the ATEC scores after intervention in experimental group were less than 20 in order to be normal, indicating that a 6-week intervention might be too short to cause autistic children to achieve optimal outcomes [42,43,47]. In the future, longer and newer progressive self–other transposition group interventions need to be developed to help autistic children achieve normality, which will prove the self–others switch system of SOME to a greater extent.

The current study divided 0–0.25 Hz into 0–0.01, 0.01–0.05, 0.05–0.1, 0.1–0.15, 0.15–0.2, and 0.2–0.25 Hz [69] and then calculated the *M* and *SD* of mAFFs among these multiple-frequency bands, but 0–0.25 Hz can be divided into other frequency bands [70,71,72,73,74,75,76,77]; then, the *M* and *SD* of mAFFs may change, which is worth exploring in the future. The best solution is to find all frequencies that exist and use the energy rank variability of them rather than dividing them into multiple-frequency bands.

The current study found that maternal average energy rank and energy rank variability among multiple-frequency bands in some brain areas were associated with the progressive self–other transposition group intervention in autistic children, which was an indirect influence. In the future, the relationship between autistic children’s average energy rank and energy rank variability and the intervention should be investigated, although it is difficult to carry out resting-state fMRI for autistic children.

## 5. Conclusions

The current study established the progressive self–other transposition group intervention to specifically improve autistic children’s self–other distinction abilities and reduce the autism symptoms and then investigated its relevant maternal brain areas. The results showed that the progressive self–other transposition group intervention successfully improved autistic children’s transposition abilities and reduced their autism symptoms; the intervention effects could be applied to daily life and last up to a month. The maternal mALFFs, average energy rank, and energy rank variability of mAFFs could predict autistic children’s transposition abilities, autism symptoms, and intervention effects with some overlap and some difference in maternal self–other distinction, sensorimotor, visual, facial expression recognition, language, memory and emotion, and self-consciousness networks, which were part of the maternal neural markers of the progressive self–other transposition group intervention effects for autistic children.

## Figures and Tables

**Figure 1 brainsci-13-00774-f001:**
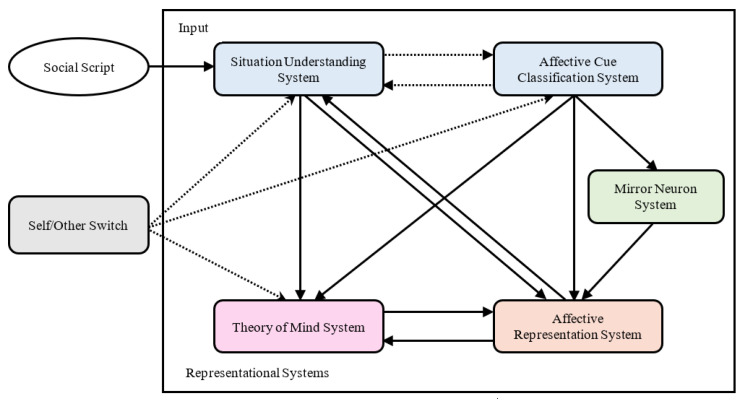
The picture of SOME [13].

**Figure 2 brainsci-13-00774-f002:**
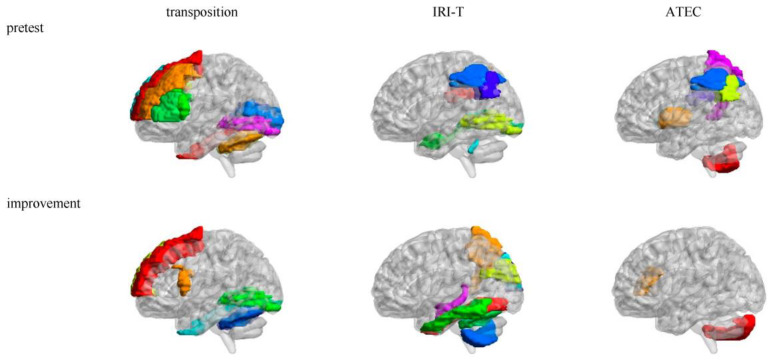
Relevant brain areas of the pretest and improvement in mALFFs. Note: The different color blocks are different AAL brain areas.

**Figure 3 brainsci-13-00774-f003:**
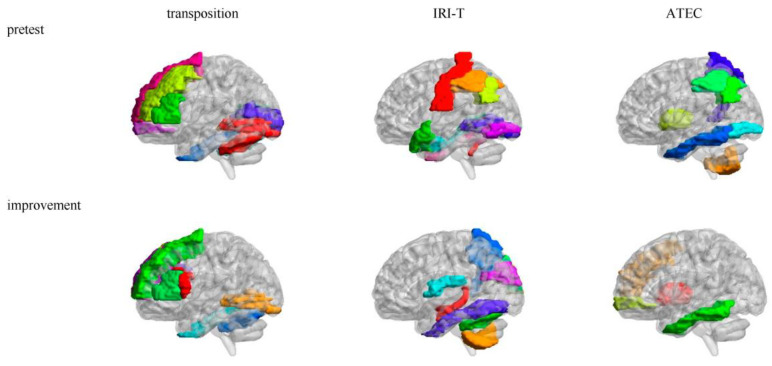
Relevant brain areas of the pretest and improvement in mAFFs’ *M.* Note: The different color blocks are different AAL brain areas.

**Figure 4 brainsci-13-00774-f004:**
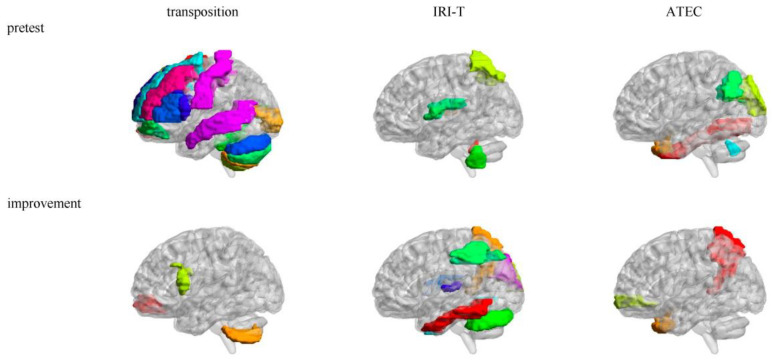
Relevant brain areas of the pretest and improvement in mAFFs’ *SD.* Note: The different color blocks are different AAL brain areas.

**Table 1 brainsci-13-00774-t001:** The progressive self–other transposition group intervention.

	Self-Centeredness (Primary Stage)	Embody Transposition (Primary and Middle Stages)	Other-Centeredness (Middle and High Stages)	Other-Expectation (High Stage)
Cognitive transposition	Carry out cognitive tasks on your side	Carry out cognitive tasks on others’ side	Guess how others carry out cognitive tasks	Guess and do what others expect you to do
Emotional transposition	Make expressions yourself	Copy others’ expressions	Guess others’ feelings without copying	Guess and do what others expect you to do to deal with their feelings
Role transposition	Play your role in a role-playing game	Play others’ roles in a role-playing game	Guess how others act with their roles in a role-playing game	Guess and do what others expect you to do with your role in a role-playing game

**Table 2 brainsci-13-00774-t002:** The indexes in the experimental and control groups and their comparison.

Test Time	Dimension	Experimental Group *M* ± *SD* (*n* = 16)	*t* (Single)	Cohen’s *d*	Control Group *M* ± *SD* (*n* = 11)	*t* (Single)	Cohen’s *d*	*t* (Independent)	Cohen’s *d*
Pretest	Three mountains	1.81 ± 1.17							
	Contingency	0.69 ± 0.87							
	Lie	2 ± 1.59							
	Transposition	1.45 ± 1.06							
	PT	14.75 ± 6.02			15.09 ± 5.58			−0.15	-
	FS	18.5 ± 6.97			15.09 ± 7.23			1.23	-
	IRI-T	33.25 ± 11.93			30.18 ± 12.05			0.65	-
	Language	9.06 ± 3.09			9.64 ± 2.29			−0.52	-
	Social interaction	17.94 ± 6.2			18.64 ± 5.26			−0.31	-
	Cognition	14.75 ± 3.49			15.45 ± 4.59			−0.45	-
	Behavior	24.75 ± 9.73			16.36 ± 7.28			2.42 *	0.98
	ATEC	66.5 ± 15.87			60.09 ± 12.55			1.12	-
Posttest	Three mountains	2.88 ± 0.34							
	Contingency	1.06 ± 1							
	Lie	3.19 ± 1.05							
	Transposition	2.29 ± 0.7							
	PT	18.38 ± 5.29			17.55 ± 6.31				
	FS	20.5 ± 6.8			15.45 ± 6.53				
	IRI-T	38.88 ± 11.56			33 ± 12.4				
	Language	8.75 ± 3.45			10 ± 3.07				
	Social interaction	17.88 ± 5.6			18.27 ± 6.86				
	Cognition	12.5 ± 5.07			15.82 ± 4.87				
	Behavior	19.44 ± 10.65			16.91 ± 8.35				
	ATEC	58.56 ± 17.87			61 ± 20.84				
Improvement	Three mountains	1.06 ± 1.12	3.78 **	0.95					
	Contingency	0.38 ± 0.72	2.09	-					
	Lie	1.19 ± 1.17	4.07 ***	1.02					
	Transposition	0.84 ± 0.56	5.98 ***	1.50					
	PT	3.63 ± 3.79	3.82 **	0.96	2.45 ± 3.8	2.14	-		
	FS	2 ± 4.87	1.64	-	0.36 ± 4.84	0.25	-		
	IRI-T	5.63 ± 7.69	2.93 **	0.73	2.82 ± 7.73	1.21	-		
	Language	0.31 ± 2.3	0.54	-	−0.36 ± 2.73	−0.44	-		
	Social interaction	0.06 ± 4.84	0.05	-	0.36 ± 4.39	0.28	-		
	Cognition	2.25 ± 3.97	2.26 *	0.57	−0.36 ± 2.2	−0.55	-		
	Behavior	5.31 ± 6.84	3.11 **	0.78	−0.55 ± 4.97	−0.36	-		
	ATEC	7.94 ± 14.52	2.19 *	0.55	−0.91 ± 10.04	−0.3	-		
Tracking improvement	PT	3.31 ± 4.72	2.81 *	0.7					
	FS	1 ± 5.62	0.71	-					
	IRI-T	4.31 ± 8.64	2	-					
	Language	1.25 ± 1.88	2.66 *	0.66					
	Social interaction	0.25 ± 4.81	0.21	-					
	Cognition	1.88 ± 3.36	2.23 *	0.56					
	Behavior	5.56 ± 5.19	4.29 ***	1.07					
	ATEC	8.94 ± 10.08	3.55 **	0.89					

Note: * *p* < 0.05; ** *p* < 0.01; *** *p* < 0.001.

**Table 3 brainsci-13-00774-t003:** Correlations of mALFFs with the pretest and improvement.

AAL Brain Area	MNI (mm) (x, y, z)	ALFFs	Transposition Pretest	Transposition Improvement	IRI-T Pretest	IRI-T Improvement	ATEC Pretest	ATEC Improvement
Frontal_Sup_L	(−18.45, 34.81, 42.2)	1 ± 0.07	0.620 *	−0.511 *				
Frontal_Mid_L	(−33.43, 32.73, 35.46)	1.05 ± 0.09	0.544 *					
Frontal_Mid_R	(37.59, 33.06, 34.04)	1.09 ± 0.08	0.523 *					
Frontal_Inf_Oper_L	(−48.43, 12.73, 19.02)	1.12 ± 0.08	0.563 *	−0.674 **				
Frontal_Inf_Tri_L	(−45.58, 29.91, 13.99)	1.14 ± 0.09	0.539 *					
Frontal_Inf_Tri_R	(50.33, 30.16, 14.17)	1.06 ± 0.06						0.510 *
Frontal_Sup_Medial_R	(9.1, 50.84, 30.22)	1.24 ± 0.14	0.550 *	−0.523 *				
Hippocampus_L	(−25.03, −20.74, −10.13)	0.96 ± 0.1				0.785 **		
ParaHippocampal_R	(25.38, −15.15, −20.47)	1.3 ± 0.12			0.536 *			
Calcarine_L	(−7.14, −78.67, 6.44)	0.77 ± 0.14	−0.539 *					
Calcarine_R	(15.99, −73.15, 9.4)	0.82 ± 0.14	−0.573 *					
Cuneus_L	(−5.93, −80.13, 27.22)	0.58 ± 0.16				−0.565 *		
Cuneus_R	(13.51, −79.36, 28.23)	0.67 ± 0.26				−0.641 **		
Lingual_L	(−14.62, −67.56, −4.63)	1.01 ± 0.14	−0.746 **	0.650 **	−0.533 *			
Lingual_R	(16.29, −66.93, −3.87)	1 ± 0.15	−0.656 **	0.501 *	−0.559 *			
Occipital_Sup_R	(24.29, −80.85, 30.59)	0.41 ± 0.14				−0.715 **		
Fusiform_L	(−31.16, −40.3, −20.23)	0.94 ± 0.06				0.524 *		
Fusiform_R	(33.97, −39.1, −20.18)	0.88 ± 0.05	−0.552 *	0.660 **				
Parietal_Inf_L	(−42.8, −45.82, 46.74)	0.77 ± 0.17			−0.621 *		0.552 *	
Parietal_Inf_R	(46.46, −46.29, 49.54)	0.8 ± 0.18			−0.507 *		0.574 *	
SupraMarginal_R	(57.61, −31.5, 34.48)	0.97 ± 0.13			−0.505 *		0.548 *	
Angular_L	(−44.14, −60.82, 35.59)	0.71 ± 0.19			−0.637 **		0.623 **	
Precuneus_L	(−7.24, −56.07, 48.01)	0.91 ± 0.16					0.525 *	
Precuneus_R	(9.98, −56.05, 43.77)	0.97 ± 0.19				−0.556 *		
Putamen_L	(−23.91, 3.86, 2.4)	0.81 ± 0.05					0.660 **	
Putamen_R	(27.78, 4.91, 2.46)	0.83 ± 0.05					0.561 *	
Pallidum_L	(−17.75, −0.03, 0.21)	0.85 ± 0.07					0.572 *	
Temporal_Inf_L	(−49.77, −28.05, −23.17)	0.85 ± 0.07				0.584 *		
Cerebelum_Crus2_R	(11.86, −69.68, −35.22)	0.61 ± 0.19						−0.521 *
Cerebelum_6_L	(−22.28, −60.11, −24.75)	0.99 ± 0.09	−0.548 *	0.559 *		0.603 *		
Cerebelum_6_R	(23.67, −59.04, −25.03)	0.99 ± 0.08	−0.591 *	0.573 *				
Cerebelum_8_L	(−22.87, −58.47, −52)	0.81 ± 0.06				0.712 **		
Cerebelum_8_R	(21.88, −60.07, −51.82)	0.8 ± 0.09					0.684 **	
Vermis_10	(−2.17, −48.85, −35.2)	1.73 ± 0.31			0.607 *			

Note: *n* = 16. * *p* < 0.05; ** *p* < 0.01.

**Table 4 brainsci-13-00774-t004:** Correlations of mAFFs’ *M* with the pretest and improvement.

AAL Brain Area	MNI (mm) (x, y, z)	mAFFs’ *M*	Transposition Pretest	Transposition Improvement	IRI-T Pretest	IRI-T Improvement	ATEC Pretest	ATEC Improvement
Frontal_Sup_L	(−18.45, 34.81, 42.2)	0.96 ± 0.06	0.500 *	−0.547 *				
Frontal_Mid_L	(−33.43, 32.73, 35.46)	0.98 ± 0.06	0.598 *					
Frontal_Mid_R	(37.59, 33.06, 34.04)	1.03 ± 0.06						0.504 *
Frontal_Inf_Oper_L	(−48.43, 12.73, 19.02)	1.06 ± 0.07		−0.651 **				
Frontal_Inf_Oper_R	(50.2, 14.98, 21.41)	1.03 ± 0.06		−0.507 *				
Frontal_Inf_Tri_L	(−45.58, 29.91, 13.99)	1.07 ± 0.07	0.591 *	−0.537 *				
Rolandic_Oper_L	(−47.16, −8.48, 13.95)	0.98 ± 0.08				0.596 *		
Frontal_Sup_Medial_L	(−4.8, 49.17, 30.89)	1.22 ± 0.11		−0.507 *				
Frontal_Sup_Medial_R	(9.1, 50.84, 30.22)	1.18 ± 0.12		−0.557 *				
Frontal_Mid_Orb_R	(8.16, 51.67, −7.13)	1.24 ± 0.18	0.511 *					0.502 *
Hippocampus_L	(−25.03, −20.74, −10.13)	0.99 ± 0.09				0.776 **		
ParaHippocampal_R	(25.38, −15.15, −20.47)	1.35 ± 0.11			0.536 *			
Calcarine_L	(−7.14, −78.67, 6.44)	0.7 ± 0.12	−0.536 *					
Calcarine_R	(15.99, −73.15, 9.4)	0.76 ± 0.12	−0.570 *					
Cuneus_L	(−5.93, −80.13, 27.22)	0.5 ± 0.14				−0.580 *		
Cuneus_R	(13.51, −79.36, 28.23)	0.6 ± 0.22				−0.631 **		
Lingual_L	(−14.62, −67.56, −4.63)	0.97 ± 0.11	−0.758 **	0.618 *	−0.515 *			
Lingual_R	(16.29, −66.93, −3.87)	0.95 ± 0.12	−0.625 **		−0.553 *			
Occipital_Sup_R	(24.29, −80.85, 30.59)	0.38 ± 0.13				−0.670 **		
Occipital_Inf_L	(−36.36, −78.29, −7.84)	0.62 ± 0.12			−0.514 *		0.531 *	
Fusiform_L	(−31.16, −40.3, −20.23)	0.97 ± 0.07				0.586 *		
Fusiform_R	(33.97, −39.1, −20.18)	0.91 ± 0.06	−0.549 *	0.535 *				
Postcentral_L	(−42.46, −22.63, 48.92)	0.84 ± 0.12			−0.506 *			
Parietal_Inf_L	(−42.8, −45.82, 46.74)	0.71 ± 0.15			−0.632 **		0.532 *	
Parietal_Inf_R	(46.46, −46.29, 49.54)	0.74 ± 0.16			−0.528 *		0.505 *	
Angular_L	(−44.14, −60.82, 35.59)	0.63 ± 0.17			−0.616 *		0.631 **	
Precuneus_L	(−7.24, −56.07, 48.01)	0.81 ± 0.14					0.509 *	
Precuneus_R	(9.98, −56.05, 43.77)	0.87 ± 0.17				−0.589 *		
Putamen_L	(−23.91, 3.86, 2.4)	0.81 ± 0.04					0.635 **	
Putamen_R	(27.78, 4.91, 2.46)	0.84 ± 0.04						0.544 *
Pallidum_L	(−17.75, −0.03, 0.21)	0.87 ± 0.06					0.711 **	
Temporal_Pole_Sup_L	(−39.88, 15.14, −20.18)	2 ± 0.28			0.576 *			
Temporal_Inf_L	(−49.77, −28.05, −23.17)	0.87 ± 0.08					0.529 *	0.501 *
Temporal_Inf_R	(53.69, −31.07, −22.32)	0.84 ± 0.09			−0.511 *			
Cerebelum_6_L	(−22.28, −60.11, −24.75)	0.99 ± 0.1	−0.499 *			0.653 **		
Cerebelum_6_R	(23.67, −59.04, −25.03)	0.96 ± 0.09	−0.567 *	0.512 *				
Cerebelum_8_L	(−22.87, −58.47, −52)	0.85 ± 0.05				0.695 **		
Cerebelum_8_R	(21.88, −60.07, −51.82)	0.83 ± 0.09					0.696 **	
Vermis_10	(−2.17, −48.85, −35.2)	1.86 ± 0.36			0.684 **			

Note: *n* = 16. * *p* < 0.05; ** *p* < 0.01.

**Table 5 brainsci-13-00774-t005:** Correlations mAFFs’ *SD* with the pretest and improvement.

AAL Brain Area	MNI (mm) (x, y, z)	mAFFs’ *SD*	Transposition Pretest	Transposition Improvement	IRI-T Pretest	IRI-T Improvement	ATEC Pretest	ATEC Improvement
Frontal_Sup_L	(−18.45, 34.81, 42.2)	0.1 ± 0.05	0.603 *					
Frontal_Sup_R	(21.9, 31.12, 43.82)	0.12 ± 0.05	0.560 *					
Frontal_Mid_L	(−33.43, 32.73, 35.46)	0.12 ± 0.06	0.560 *					
Frontal_Mid_R	(37.59, 33.06, 34.04)	0.13 ± 0.05	0.519 *					
Frontal_Mid_Orb_L	(−30.65, 50.43, −9.62)	0.16 ± 0.11	0.606 *					
Frontal_Mid_Orb_R	(33.18, 52.59, −10.73)	0.16 ± 0.1	0.723 **	−0.548 *				
Frontal_Inf_Oper_L	(−48.43, 12.73, 19.02)	0.09 ± 0.03	0.576 *	−0.498 *				
Frontal_Inf_Tri_L	(−45.58, 29.91, 13.99)	0.12 ± 0.04	0.577 *					
Rolandic_Oper_L	(−47.16, −8.48, 13.95)	0.04 ± 0.02			−0.525 *			
Rolandic_Oper_R	(52.65, −6.25, 14.63)	0.04 ± 0.02				0.549 *		
Frontal_Sup_Medial_L	(−4.8, 49.17, 30.89)	0.15 ± 0.06	0.505 *					
Frontal_Sup_Medial_R	(9.1, 50.84, 30.22)	0.15 ± 0.06	0.537 *					
Frontal_Mid_Orb_R	(8.16, 51.67, −7.13)	0.22 ± 0.18						0.505 *
Calcarine_L	(−7.14, −78.67, 6.44)	0.1 ± 0.05	−0.544 *					
Calcarine_R	(15.99, −73.15, 9.4)	0.08 ± 0.05	−0.608 *					
Cuneus_L	(−5.93, −80.13, 27.22)	0.11 ± 0.06					0.525 *	
Cuneus_R	(13.51, −79.36, 28.23)	0.11 ± 0.07				−0.517 *		
Occipital_Sup_L	(−16.54, −84.26, 28.17)	0.03 ± 0.02					0.526 *	
Occipital_Sup_R	(24.29, −80.85, 30.59)	0.04 ± 0.02				−0.589 *		
Fusiform_R	(33.97, −39.1, −20.18)	0.05 ± 0.02					0.766 **	
Postcentral_L	(−42.46, −22.63, 48.92)	0.08 ± 0.05	−0.503 *					
Parietal_Sup_L	(−23.45, −59.56, 58.96)	0.07 ± 0.05			−0.553 *			
Parietal_Inf_L	(−42.8, −45.82, 46.74)	0.08 ± 0.04				0.577 *		
Angular_L	(−44.14, −60.82, 35.59)	0.09 ± 0.04					0.521 *	
Precuneus_R	(9.98, −56.05, 43.77)	0.14 ± 0.04				−0.660 **		−0.561 *
Heschl_L	(−41.99, −18.88, 9.98)	0.14 ± 0.09				0.529 *		
Heschl_R	(45.86, −17.15, 10.41)	0.15 ± 0.05			0.652 **			
Temporal_Mid_L	(−55.52, −33.8, −2.2)	0.06 ± 0.04	−0.571 *					
Temporal_Pole_Mid_R	(44.22, 14.55, −32.23)	0.09 ± 0.03					0.629 **	0.546 *
Temporal_Inf_L	(−49.77, −28.05, −23.17)	0.06 ± 0.03				0.766 **		
Temporal_Inf_R	(53.69, −31.07, −22.32)	0.04 ± 0.02				−0.552 *		
Cerebelum_Crus1_L	(−30.79, −71.02, −35.89)	0.1 ± 0.08	−0.516 *			0.581 *		
Cerebelum_Crus2_L	(−8.92, −71.02, −35.89)	0.05 ± 0.03	−0.524 *					
Cerebelum_4_5_L	(−15.4, −46.41, −19.89)	0.18 ± 0.1	0.536 *					
Cerebelum_7b_L	(−27.77, −64.64, −49.49)	0.07 ± 0.03	−0.685 **	0.546 *				
Cerebelum_8_L	(−22.87, −58.47, −52)	0.08 ± 0.02	−0.623 **					
Cerebelum_9_L	(−9.96, −52.62, −51.13)	0.18 ± 0.06			0.538 *			
Vermis_8	(−1.44, −67.4, −38.72)	0.08 ± 0.03					0.509 *	
Vermis_10	(−2.17, −48.85, −35.2)	0.36 ± 0.13			0.679 **			

Note: *n* = 16. * *p* < 0.05; ** *p* < 0.01.

**Table 6 brainsci-13-00774-t006:** (a) Consistency between mAFFs’ *M* and mALFFs in pretests; (b) consistency between mAFFs’ *M* and mALFFs in improvements.

(a)					
AAL	Transposition Pretest	AAL	IRI-T Pretest	AAL	ATEC Pretest
Frontal_Sup_L	3	ParaHippocampal_R	3	Parietal_Inf_L	3
Frontal_Mid_L	3	Lingual_L	3	Parietal_Inf_R	3
Frontal_Inf_Tri_L	3	Lingual_R	3	Angular_L	3
Calcarine_L	3	Parietal_Inf_L	3	Precuneus_L	3
Calcarine_R	3	Parietal_Inf_R	3	Putamen_L	3
Lingual_L	3	Angular_L	3	Pallidum_L	3
Lingual_R	3	Vermis_10	3	Cerebelum_8_R	3
Fusiform_R	3	Occipital_Inf_L	2	Occipital_Inf_L	2
Cerebelum_6_L	3	Postcentral_L	2	Temporal_Inf_L	2
Cerebelum_6_R	3	Temporal_Pole_Sup_L	2	SupraMarginal_R	1
Frontal_Mid_Orb_R	2	Temporal_Inf_R	2	Putamen_R	1
Frontal_Mid_R	1	SupraMarginal_R	1		
Frontal_Inf_Oper_L	1				
Frontal_Sup_Medial_R	1				
**(b)**					
**AAL**	**Transposition** **Improvement**	**AAL**	**IRI-T** **Improvement**	**AAL**	**ATEC** **Improvement**
Frontal_Sup_L	3	Hippocampus_L	3	Frontal_Mid_R	2
Frontal_Inf_Oper_L	3	Cuneus_L	3	Frontal_Mid_Orb_R	2
Frontal_Sup_Medial_R	3	Cuneus_R	3	Putamen_R	2
Lingual_L	3	Occipital_Sup_R	3	Temporal_Inf_L	2
Fusiform_R	3	Fusiform_L	3	Frontal_Inf_Tri_R	1
Cerebelum_6_R	3	Precuneus_R	3	Cerebelum_Crus2_R	1
Frontal_Inf_Oper_R	2	Cerebelum_6_L	3		
Frontal_Inf_Tri_L	2	Cerebelum_8_L	3		
Frontal_Sup_Medial_L	2	Rolandic_Oper_L	2		
Lingual_R	1	Temporal_Inf_L	1		
Cerebelum_6_L	1				

Note: In the table, “3” denotes the common relevant brain areas both in mAFFs’ *M* and mALFFs; “2” denotes the relevant brain areas only in mAFFs’ *M*; and “1” denotes the relevant brain areas only in mALFFs.

**Table 7 brainsci-13-00774-t007:** (a) Consistency between mAFFs’ *SD* and mALFFs in pretests; (b) consistency between mAFFs’ *SD* and mALFFs in improvements.

(a)					
AAL	Transposition Pretest	AAL	IRI-T Pretest	AAL	ATEC Pretest
Frontal_Sup_L	3	Vermis_10	3	Angular_L	3
Frontal_Mid_L	3	Rolandic_Oper_L	2	Cuneus_L	2
Frontal_Mid_R	3	Parietal_Sup_L	2	Occipital_Sup_L	2
Frontal_Inf_Oper_L	3	Heschl_R	2	Fusiform_R	2
Frontal_Inf_Tri_L	3	Cerebelum_9_L	2	Temporal_Pole_Mid_R	2
Frontal_Sup_Medial_R	3	ParaHippocampal_R	1	Vermis_8	2
Calcarine_L	3	Lingual_L	1	Parietal_Inf_L	1
Calcarine_R	3	Lingual_R	1	Parietal_Inf_R	1
Frontal_Sup_R	2	Parietal_Inf_L	1	SupraMarginal_R	1
Frontal_Mid_Orb_L	2	Parietal_Inf_R	1	Precuneus_L	1
Frontal_Mid_Orb_R	2	SupraMarginal_R	1	Putamen_L	1
Frontal_Sup_Medial_L	2	Angular_L	1	Putamen_R	1
Postcentral_L	2			Pallidum_L	1
Temporal_Mid_L	2			Cerebelum_8_R	1
Cerebelum_Crus1_L	2				
Cerebelum_Crus2_L	2				
Cerebelum_4_5_L	2				
Cerebelum_7b_L	2				
Cerebelum_8_L	2				
Lingual_L	1				
Lingual_R	1				
Fusiform_R	1				
Cerebelum_6_L	1				
Cerebelum_6_R	1				
**(b)**					
**AAL**	**Transposition** **Improvement**	**AAL**	**IRI-T** **Improvement**	**AAL**	**ATEC** **Improvement**
Frontal_Inf_Oper_L	3	Cuneus_R	3	Frontal_Mid_Orb_R	2
Frontal_Mid_Orb_R	2	Occipital_Sup_R	3	Precuneus_R	2
Cerebelum_7b_L	2	Precuneus_R	3	Temporal_Pole_Mid_R	2
Frontal_Sup_L	1	Temporal_Inf_L	3	Frontal_Inf_Tri_R	1
Frontal_Sup_Medial_R	1	Rolandic_Oper_R	2	Cerebelum_Crus2_R	1
Lingual_L	1	Parietal_Inf_L	2		
Lingual_R	1	Heschl_L	2		
Fusiform_R	1	Temporal_Inf_R	2		
Cerebelum_6_L	1	Cerebelum_Crus1_L	2		
Cerebelum_6_R	1	Hippocampus_L	1		
		Cuneus_L	1		
		Fusiform_L	1		
		Cerebelum_6_L	1		
		Cerebelum_8_L	1		

Note: In the table, “3” denotes the common relevant brain areas both in mAFFs’ *SD* and mALFFs; “2” denotes the relevant brain areas only in mAFFs’ *SD*; and “1” denotes the relevant brain areas only in mALFFs.

## Data Availability

The datasets generated during and/or analyzed during the current study are available from the corresponding author on reasonable request.

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
