# Peer review of "Maternal Variability of Amplitudes of Frequency Fluctuations Is Related to the Progressive Self–Other Transposition Group Intervention in Autistic Children"

_brainsci, 2023, doi:10.3390/brainsci13050774_

Round 1
Reviewer 1 Report
1. The abstract should be broadened to give additional quantitative results.
2. At the end of your abstract, please provide a "take-home" message.
3. Rearrange keywords alphabetically.
4. It is encouraged not used abbreviations in the keywords section.
5. The Reviewer do not see the novel in the present article. My examination revealed that several similar previous publications appear to appropriately address the issues you have brought up in the current submission. Please emphasize it more advance in the introduction section if there are any more truly something really new.
6. Previous study related needs to explain in the introduction section consisting of their work, their novelty, and their limitations to show the research gaps that intend to be filled in the present study.
7. Previous study related to autism children treatment need to be explained. One of the is from Afif et al that using sensory hug machine. The introduction and/or discussion part of an article should contain this crucial information. In addition, to support this explanation, the MDPI-suggested reference should be included as follows: Afif, I. Y.; Manik, A. R.; Munthe, K.; Maula, M. I.; Ammarullah, M. I.; Jamari, J.; Winarni, T. I. Physiological Effect of Deep Pressure in Reducing Anxiety of Children with ASD during Traveling: A Public Transportation Setting. Bioengineering 2022, 9, 157. https://doi.org/10.3390/bioengineering9040157
8. Rather than relying just on the predominate text as it already exists, the authors could incorporate more illustrations as figures in the materials and methods section that illustrate the workflow of the current study.
9. What is the baseline of patient selection? Is there any protocol, standard, or basis that has been followed? It is unclear since the patient is very heterogeneous with a small number. The resonance involved impacts the present result makes this study flaws. One major reason for rejecting this paper.
10. It's also important to provide more particular information on tools, such as the manufacturer, the country, and the specification.
11. The inaccuracy and tolerance of the experimental equipment used in this inquiry are critical details that must be included in the article.
12. Results must be compared to similar past research.
13. Overall, discussion in the present article is extremely poor. The Authors must extend their discussion and make a comprehensive explanation. Just not simply mention the results with brief explanation.
14. Please include the limitation of the present study, it is missing.
15. Provide a paragraph-length conclusion rather than the present form's point-by-point description.
16. The conclusion section needs to explain further research.
17. The reference needs to be enriched from the literature published five years back. MDPI reference is strongly recommended.
18. In the whole of the manuscript, the authors sometimes made a paragraph only consisting of one or two sentences that made the explanation not clearly understood. The authors need to extend their explanation to become a more comprehensive paragraph. In one paragraph, it is recommended to consist of at least 3 sentences with 1 sentence as the main sentence and the other sentences as supporting sentences.
19. The authors were encouraged to proofread their work due to grammatical problems and linguistic style.
20. After revision, provide a graphical abstract for submission.
Author Response
Thank you very much for your enlightening and meticulous comments! We have improved the quality of the article in the revised manuscript according to your suggestions. Please see the blue text for the revision. Please refer to the attachment for our reply.

Reviewer 2 Report
Dear Editor, thank you for inviting me to review the paper entitled “The relevant maternal brain activity of the progressive self/other transposition group intervention in ASD children” for publication on “Brain Sciences”.
The authors implemented an empathy-based intervention program for autistic children and evaluated treatment efficacy in a sample of N = 16 children compared to a N = 12 control group with as-usual intervention. The authors found significant changes in the intervention groups and several correlations between maternal brain functioning and outcome variables. The research work, as well as the theoretical framework are interesting, as well as the effort to evaluate mechanisms of change, the role of maternal brain functioning, and process measures. However, I have several concerns that prevent me from recommending the paper for publication in the present form.
In summary:
-
The introduction lacks relevant literature
-
The research questions and the hypothesis are completely missing
-
It is not clear which are the outcome variables with respect to the process measures
-
Sampling procedure is unspecified
-
Diagnostic procedure for ASD is unspecified
-
Children are characterized only by fluid intelligence and no developmental measure is reported and symptom severity is not taken into account for baseline comparisons
-
Measures are not clearly described
-
Tested variables are highly correlated and multicollinearity is not addressed
-
Multiple comparisons are not handled and no correction was applied
-
Inferential tests are not applied to detect significant differences between the two groups pre- and post- treatment
-
Results are confusing and difficult to follow
-
Limitations are missing
As a general comment, the paper is difficult to understand, to read, and to follow. The English should be extensively revised since there are several sentences which are unclear.
The title itself is confusing, I suggest rephrasing.
Abstract:
-
In general the abstract is unclear, methods are not specified, the study target is unclear, the outcome variables are not specified, and the link between maternal brain activation and the study objective is missing
Introduction:
-
Line 33: reference missing. In general, several sentences lack references and research support. In many other cases, the research literature cited is partial or incomplete.
-
Adhere to APA definition of Autism and Autism Spectrum Disorder (ASD)
-
Literature on intervention effects is missing and partial
-
The language used is not in line with recent guidelines that recommend identity-first language and the use of ASD limited to diagnostic criteria and measures. As well, the framework of neurodiversity implies “treat impairment, not diversity” focusing on specific variables, e.g., developmental delay, cognitive impairment, self-injurious behavior, etc., and not on Autism as a broader condition. Leadbitter, K., Buckle, K. L., Ellis, C., & Dekker, M. (2021). See: Autistic Self-Advocacy and the Neurodiversity Movement: Implications for Autism Early Intervention Research and Practice. Frontiers in psychology, 12, 635690. https://doi.org/10.3389/fpsyg.2021.635690, and Botha, M., Hanlon, J., & Williams, G. L. (2021). Does language matter? Identity-first versus person-first language use in autism research: A response to Vivanti. Journal of Autism and Developmental Disorders, 1-9.
-
Arguments about intervention efficacy are definitely exaggerated and the cited literature does not support them. Only an unclear minority of children eventually exit the spectrum and do not satisfy diagnostic criteria after Autism treatment, especially children whose diagnosis was already nuanced, without cognitive impairment and, in general, with high levels of functioning. See: Helt, M., Kelley, E., Kinsbourne, M., Pandey, J., Boorstein, H., Herbert, M., & Fein, D. (2008). Can children with autism recover? If so, how?. Neuropsychology review, 18(4), 339–366. https://doi.org/10.1007/s11065-008-9075-9, and Fein, D., Barton, M., Eigsti, I. M., Kelley, E., Naigles, L., Schultz, R. T., Stevens, M., Helt, M., Orinstein, A., Rosenthal, M., Troyb, E., & Tyson, K. (2013). Optimal outcome in individuals with a history of autism. Journal of child psychology and psychiatry, and allied disciplines, 54(2), 195–205. https://doi.org/10.1111/jcpp.12037
-
Even literature on super-responders did not show such an optimistic picture in the general sense. Please see: Sandbank, M., Bottema-Beutel, K., Crowley, S., Cassidy, M., Dunham, K., Feldman, J. I., Crank, J., Albarran, S. A., Raj, S., Mahbub, P., & Woynaroski, T. G. (2020). Project AIM: Autism intervention meta-analysis for studies of young children. Psychological bulletin, 146(1), 1–29. https://doi.org/10.1037/bul0000215. Please widen the literature, reduce the strength of arguments and include more generalized evidence, as well as reviews and meta-analysis of efficacy and treatment effect size.
-
Line 45: A diagram representing the internal logic of the SOME model would greatly help to understand the theoretical model.
-
Line 60: This is a causal evidence that need strong research literature to support it
-
Line 66: Poor ref support
-
Line 76: English is incorrect, the sentence is unclear and lacks research literature support
-
Line 86: English
-
Line 96-98: What does this sentence mean?
-
Line 99-102: Operational definition is missing
-
Line 114-116: Unclear sentence
Methods:
-
Line 122: unit of measure missing
-
No information about sampling procedure
-
Line 133: which version?
-
Line 143: what do the authors mean by pseudo-random?
-
Was the effect of order controlled? Since outcome variables are highly correlated, this is even more important
-
No clear description and examples of intervention tasks and treatment setting
-
Line 162: what does this mean?
-
Line 169: what does this mean?
-
Line 182-183: details on EFA are needed
-
Test-retest reliability reliability is missing. Are (and why are) these measures suitable as outcome measures?
-
The diagnostic procedure is missing. Who made the diagnosis? Following which criteria? Was a gold-standard instrument, e.g., the ADOS-2, used to confirm the diagnosis?
-
The instruments used are poorly described in their psychometric properties
-
The outcome variables are unclear as well as the links with the theoretical framework. I had the impression that process variables and outcome variables are sometimes confused
-
Were the data checked for the assumptions? Parametric inferential tests and analysis of variance have a set of ground hypothesis that need to be satisfied to be applied
-
Inferential tests should be performed between the two samples both pre- and post- treatment to test for significant differences.
-
Line 213-214: what does this mean?
-
Line 232: Reference needed
-
Line 233: justify this argument
-
Line 239: with which method?
-
Line 246: AAL needs details and refs
-
Lines 252-253: Why would the averaging of the different areas compensate for the need for multiple comparisons? No statistical justification is provided. Moreover, the multiple comparison problem concerns the outcome variables that are tested for correlation multiple times with all the brain activation variables coming from the different areas. This analysis definitely requires control for multiple comparisons.
Results:
-
Correlations should be performed also post treatment
-
Correlations should be represented on a correlation plot for clarity
-
The tables are completely unreadable
-
Results are hardly readable
-
Correlations are often shared between different brain areas. Correlations between the activation of different brain areas were not tested (and should be) and may be probably high. To what extent can we test the specificity of these results? Why did the authors not use predictive modeling through linear regression or ANCOVA to take into account possible covariates / interdependence / multicollinearity? Given the reduced sample size, why did the authors not perform a PCA with 2 or 3 dimensions, verify the explained variance, and try to predict improvements in outcome measures for the experimental group?
Discussion:
-
It remains unclear how the follow-up was tested
-
The limitation part is completely missing
Author Response

(The authors gave the same response as above.)

Round 2
Reviewer 1 Report
There are following comments that needs to addressed in this stage.
1. The reviewer suggests revising the keywords without using abbreviations.
2. In the introduction section, it would improve the quality of the present work by providing an additional related figure.
3. Incorporated the suggested reference in the introduction section about studies related to ASD as follows: Effect of Short-Term Deep-Pressure Portable Seat on Behavioral and Biological Stress in Children with Autism Spectrum Disorders: A Pilot Study. Bioengineering 2022, 9, 48. https://doi.org/10.3390/bioengineering9020048
4. I order to reduce the self-citation level, the authors recommend not use their previous work as a reference over.
Author Response
1. Thank you very much for your further advices! We revised the manuscript in blue font. The keywords were modified without using abbreviations.
2. Figure 1 has been redrawn clearly in the introduction section.
3. Thank you for the reference. It was added in the introduction section.
4. Thank you for the reminding. We only kept two necessary articles of our previous works as references to explain why a multiple comparisons correction was unnecessary and could not be made for the correlation analyses in the current study. They were as follows:
Zhang, J. X., & Liu, D. Z. (2021). The gradual subjective consciousness fluctuation in implicit sequence learning and its relevant brain activity. Neuropsychologia(2), 107948.
Zhang, J. X., Yin, M., Shu, D. M., & Liu, D. Z. (2022). The establishment of the general microexpression recognition ability and its relevant brain activity. Frontiers in Human Neuroscience, 12, 894702.
Other articles with the family name “Zhang” were not any author of the current study.